# Design and Application of Phase-Only Diffractive Optical Element Based on Non-Iterative Method

Kuo Shi [ID] and Gongjian Zhang *[ID]

Department of Opto-Electronic System Engineering, Chitose Institute of Science and Technology, Chitose 066-8655, Japan
* Correspondence: zhang@photon.chitose.ac.jp

**Abstract:** In this study, we devised a method for the design of continuous phase-only holographic masks that map laser light to arbitrary target illumination patterns, which have a wide range of applications. In this method, the discrete gradient of a holographic mask is obtained by combining geometric optics and the linear assignment problem (LAP) methods, and then the entire problem is transformed into an integral problem with a discrete gradient. Finally, the least squares method is used to solve the gradient integral to complete the construction of a phase holographic mask. Due to its good continuity, this mask design method can also be applied to the production of diffractive optical elements. We discussed the effectiveness of this method by constructing two holographic masks with uniform illumination. At the same time, we successfully constructed an Einstein face holographic mask with non-uniform illumination using the LAP method for the first time. It is believed that this method can be widely used in illumination mode, ion capture and other directions.

**Keywords:** linear assignment problem (LAP); phase-only freeform; computer-generated holography

## 1. Introduction

Computer-generated holography (CGH) is a powerful three-dimensional display technology that is useful in various applications, such as in laser-beam shaping and focusing, the formation of complex images in augmented reality systems and holographic displays, structural lighting, optical traps and microscopes. [1–5]. Compared with amplitude holograms, reconstructed computer-generated holograms are brighter and they can be displayed with a single-phase spatial light modulator (SLM), which simplifies holographic display systems [6]. At present, iterative Fourier transform algorithm (IFTA) methods are widely used in the generation of phase-only holography and have achieved great success [7,8]. However, the diffractive optical element (DOE) microreliefs obtained by it are usually complex and irregular in shape, often similar to white noise. Such relief is difficult to manufacture, and the manufacturing error leads to the uncontrolled scattering of light on the microrelief and the formation of speckle structure, which greatly damages the quality of the generated light irradiation distribution. Therefore, in this study, we used a mapping illumination method to realize the design of phase-only freeform surface holograms.

Illumination patterns were originally designed based on non-imaging optics, which is a key branch of geometric optics that is developing rapidly due to extensive research in solar energy and light-emitting diode technology. An inverse problem of non-imaging optics is the design of freeform surface optical elements with a certain irradiance distribution and wavefront. In most cases, this inverse problem can be reduced to an elliptic non-linear partial differential equation (PDE). Design methods that can be used for obtaining direct numerical solutions for elliptic non-linear PDEs have only appeared in recent years. For example, in 2013, Wu et al. [9] transformed the PDE representing a beam-shaping problem into a non-linear boundary problem of the elliptical Monge–Ampère equation, based on the concept of an optimal mass transport problem. In this method, after establishing the boundary conditions of the mapping, a nine-point difference scheme is used to discretize the

PDE, and then Newton's method is used to solve the Monge–Ampère equation. However, this kind of iterative method for solving PDEs is a highly computationally complex way to solve non-linear equations, as the selection of an initial iterative value affects the probability that an equation will be solved, such that improper selection of an initial iterative value often leads to no convergent solution.

The supporting quadric method (SQM) was devised by Oliker and co-workers [10] and is widely used for designing illumination patterns; that is, for designing the optical elements of refracting mirrors to produce discrete intensity or illumination patterns at a given position [10–12]. In the SQM, a given irradiance distribution is defined as a discrete distribution approximation over a finite set of N points, and an optical surface is then represented as a set of quadric segments with N segments, with each focusing the incident beam onto one of these N points. Paraboloids, ellipsoids or hyperboloids can be used as quadric segments, depending on the problem to be solved. The parameters of quadric segments are calculated by the iterative method. The SQM has been used for the design of freeform surface lenses capable of focused illumination and imaging of light fields in a specific area.

The key challenge in the Monge–Ampère method and the SQM is to solve the mapping relationship of beams. Doskolovich et al. have conducted extensive research in this area, including on the SQM, variational methods, linear assignment problem (LAP) methods and hybrid algorithms [12–15]. In their research on the design of variational methods, they highlighted the effectiveness of an LAP approach and used iterative methods to verify a uniform lighting problem [13,14].

Different from the literature [13,14], in the current study, a non-iterative method was used to design phase-only holographic diffraction masks for several uniformly focused images and some non-uniformly focused images via an LAP method, and the influence of lens sag on final imaging quality was examined. In this non-iterative method, an optimal mass-transportation problem is transformed into an LAP that is processed by the Hungarian algorithm. Subsequently, the phase function is calculated by the gradient integration method [16–19]. This method efficiently generates a smooth phase-only diffraction mask, thereby affording a phase diffraction element that is easy to process. We demonstrated the effectiveness of this method by using it to generate holographic masks whose diffraction intensity distributions were a geometric pattern composed of four rectangles with rotational symmetry and the characters "CIST" without symmetry, and an image with an illumination distribution. To the best of our knowledge, the LAP method for focusing imaging processes with intensity distributions has rarely been reported. In view of this, in this paper, we use the LAP method to realize the design of a phase-only hologram of a non-uniformly illuminated portrait.

## 2. Design Method

### 2.1. Phase Gradient Calculation

The design problem of the phase-only holographic mask is actually a calculation of the phase function of a light field on an input plane under the condition of producing a given irradiance distribution on a given plane. In geometric optical approximations, it is common to deal with such a problem by using an eikonal function instead of a phase function. Therefore, we define a light field distribution $E(x,y) = \sqrt{I_0} \exp(\mathrm{i}kz_0)$ at $z = 0$, where the wavenumber $k = 2\pi/\lambda$, $\lambda$ is the wavelength of the beam, $(x,y)$ are the Cartesian coordinates at $z = 0$, $I_0$ is the beam intensity distribution at the illumination region $S_1$ at $z = 0$ and $z_0(x,y)$ is the eikonal function at $z = 0$. The position vector on the wavefront is $z_0 = (x, y, z_0)$. When the beam propagates to $z = f$, its light field distribution is denoted as $E(u,v) = \sqrt{I_f} \exp\left(\mathrm{i}kz_f\right)$, where $(u,v)$ is the Cartesian coordinate at $z = f$, and $z_f(u,v)$ is the optical path function at $z = f$. Let the position vector on the observation plane be

$z_f = (u, v, z_f)$. $I_f$ is the beam intensity distribution in the illumination region $S_2$ at $z = f$. By using **r** to represent the unit direction vector of $(z_f - z_0)$, it can be found that

$$\mathbf{r} = \left( u - x, v - y, z_f - z_0 \right) / L, \tag{1}$$

where $L^2 = (x - u)^2 + (y - v)^2 + (z_0 - z_f)^2$. As the light field propagates in a uniform medium, **r** is also the normal vector of $z_0$, so

$$\begin{cases} \frac{\partial z_0}{\partial x} = (u - x)/\left(z_f - z_0\right) \\ \frac{\partial z_0}{\partial y} = (v - y)/\left(z_f - z_0\right) \end{cases} \tag{2}$$

can be obtained, in which $z_f - z_0$ can be approximated by $f$. Therefore,

$$\begin{cases} \frac{\partial z_0}{\partial x} = \frac{u-x}{f} \\ \frac{\partial z_0}{\partial y} = \frac{v-y\prime}{f} \end{cases} \tag{3}$$

$$\begin{cases} u = x + \frac{\partial z_0}{\partial x} f \\ v = y + \frac{\partial z_0}{\partial y} f \end{cases}. \tag{4}$$

Equation (4) represents the coordinate correspondence of each discrete point on the $z = 0$ plane and $z = f$ plane. According to the coordinate transformation relation and the energy conservation law,

$$J(S_2)E(x,y) = E(u,v) \tag{5}$$

$$\iint_{S_1} E(x,y)dxdy = \iint_{S_2} E(u,v)dudv \tag{6}$$

can be obtained, where $J(S_2) = \partial u/\partial x \cdot \partial v/\partial y - \partial u/\partial y \cdot \partial v/\partial x$, the Jacobian matrix of the coordinate transformation relation defined in Equation (4). Equation (6) is the integral expression of energy conservation; its discrete form is

$$\sum_{n=1}^{N} |E(x_n, y_n)| = \sum_{m=1}^{M} |E(u_m, v_m)|, \tag{7}$$

where $N$ is the number of discrete points in region $S_1$, and $M$ is the number of discrete points in region $S_2$. When the light intensity distribution on the initial plane is uniform (that is, $I_0$ is constant), the following correspondence exists

$$I_f(u_m, v_m) = \sum_{n=1}^{N_m} I_0(x_n, y_n), N_m = \frac{I_f(u_m, v_m)}{I_0}, \tag{8}$$

where $N_1 + N_2 + \ldots + N_M = N$. That is, point $I_f (u_m, v_m)$ corresponds to $N_m$ mappings, and the distance of the mapping corresponding to a single point on the $S_2$ plane can be expressed as

$$D_m = \sum_{n=1}^{N_m} L(x_n, y_n, u(x_n), v(y_n)). \tag{9}$$

The total distance of the whole mapping is as follows:

$$T = \sum_{m=1}^{M} D_m. \tag{10}$$

It can be seen that the mass-transportation problem is transformed into an LAP when T is minimized. In this study, we used the Hungarian algorithm to solve the above optimization problem and found that it can easily obtain the corresponding relationship between the initial coordinate system and the target coordinate system. Thus, by substituting the obtained correspondence into Equation (3), the gradient $(f_x, f_y) = (\partial z_0/\partial x, \partial z_0/\partial y)$ on the phase surface $z_0$ can be determined. Next, we used a non-iterative gradient integration method to reconstruct the phase surface.

### 2.2. Gradient Integration Methods

The region method is a gradient integration method that uses the linear combination of gradient values in a region near the target point to represent height difference and determines the relative position of each height point by solving linear equations. For example, a least squares algorithm based on Taylor's theorem that reconstructs a wavefront under any combination of tilt, defocus and astigmatism. The relationship between the height value and the gradient can be expressed as

$$\begin{bmatrix} P_x \\ P_y \end{bmatrix} Z_0 = \begin{bmatrix} F_x \\ F_y \end{bmatrix}, \tag{11}$$

where $Z_0$ is the vector representation of the phase distribution $z_0$, $P_x$ and $P_y$ are the coefficient matrices of the combination of height points corresponding to the difference operator and $F_x$ and $F_y$ are the difference vectors of row direction and column direction, respectively, estimated according to gradient points. Based on a Taylor expansion, the integral formula of $z_0$ in the $x$-direction and $y$-direction can be approximately expressed by the following formula:

$$\begin{cases} z_{0(x,y+1)} - z_{0(x,y)} = 0.5 \times \Delta \times \left( f_{x(x,y+1)} + f_{x(x,y)} \right) \\ z_{0(x+1,y)} - z_{0(x,y)} = 0.5 \times \Delta \times \left( f_{y(x+1,y)} + f_{x(x,y)} \right) \end{cases}. \tag{12}$$

The relationship between $F_x$, $F_y$ and gradient $f_x$, $f_y$ can be expressed as follows: $F_x$ $(x, y) = (f_x (x, y + 1) + f_x (x, y)) \times 0.5\Delta$, $F_y (x, y) = (f_y (x + 1, y) + f_y (x, y)) \times 0.5\Delta$, where $\Delta = 12.5$ µm is the size of the mask pixel. Equation (12) shows that the coefficient matrices $P_x$ and $P_y$ are sparse matrices containing only the elements 1 and $-1$. The solution of the phase distribution can be obtained from the following least squares algorithm:

$$Z_0 = \left( \begin{bmatrix} P_x \\ P_y \end{bmatrix}^T \begin{bmatrix} P_x \\ P_y \end{bmatrix} \right)^{-1} \begin{bmatrix} P_x \\ P_y \end{bmatrix}^T \begin{bmatrix} F_x \\ F_y \end{bmatrix}. \tag{13}$$

## 3. Design Examples

### 3.1. Uniform Illumination

The phase holographic masks of two special images with uniform illumination were calculated at different propagation distances f, and their diffraction spectra were simulated and observed experimentally. As described in Ref. [13], the process of ray mapping can be seen as a process of focusing a beam to a specified area. The focal plane is the plane $z = f$. Therefore, the propagation distance f is also described as the focal length of the mask in the later description of this paper. Their diffraction intensity profiles were a rectangle and the characters "CIST", respectively. The details of the parameters of the intensity distribution graphs are as follows. The first image (Figure 1a) was composed of four rectangles with rotational symmetry. The total number of discrete points contained in this image was $N = 64 \times 64$, and the number of discrete points in the effective illumination area (i.e., the non-zero area of the whole image) was $M = 256$. The second image (Figure 1b) contained the characters "CIST". The total number of discrete points contained in this image was $N = 90 \times 90$, and the number of discrete points in the effective lighting area was $M = 900$. To verify the mask diffraction effect on an SLM, we set the pixel size of the image to be consistent with the pixel size of the SLM that we used, and its length and width were $\Delta = 12.5$ µm. The total times required to compute the phase functions for these images on a desktop computer (equipped with an I9-9900 core CPU @ 3.10 GHz) were approximately 80 s (Figure 1a) and 550 s (Figure 1b), respectively.

$$E(u,v) = F^{-1} \left\{ F[E_0(x,y)] \exp\left( i\frac{2\pi}{\lambda} z_T \sqrt{1 - (\lambda f_x)^2 - (\lambda f_y)^2} \right) \right\} \tag{14}$$

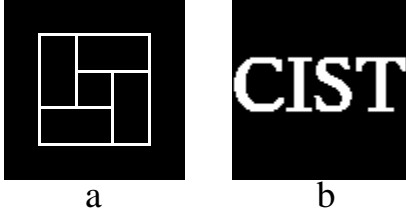

**Figure 1.** (**a**) Rectangular target irradiance distributions, (**b**) the characters "CIST" target irradiance distributions.

The angular spectrum method was used to simulate the diffraction of a hologram [20], and the diffraction spectrum calculation formula is shown in Equation (14), where $E_0$ is the angular spectrum, $(fx, fy)$ is the Fourier spatial frequency domain, $z$ is the propagation distance and $z_T$ is the propagation distance and is equal to the focal length $f$ of a mask. Equation (14) is an exact expression similar to Kirchhoff's formula. The Fresnel diffraction integral is obtained by the Fresnel approximation simplification and is not different from the exact solution. This conclusion can be proved by the stationary phase method.

Figures 2a and 3a show the calculation results of holographic masks and the simulated diffraction spectra when $f$ = 5 cm. It is well known that the diffraction of a beam after passing through this kind of mask can be regarded as a focusing process with focal length $f$. As the focal length $f$ = 5 cm was long, the curvatures of the masks were very small. The overall details of the masks were thus compressed, which made the diffraction results very fuzzy. We therefore reduced $f$ to 2 cm and re-performed simulations. As shown in Figures 2b and 3b, the diffraction result with $f$ = 2 cm was significantly better than that with $f$ = 5 cm. The two masks were of 64 × 64 pixels and 90 × 90 pixels, respectively, and thus as the $f$ value was further decreased, the curvature of the masks rapidly increased beyond the range that could be expressed by the number of pixels, resulting in large distortions. In particular, when $f$ = 1 cm, the whole pattern was completely distorted, indicating that the masks could not be optimized by continually reducing the $f$ value.

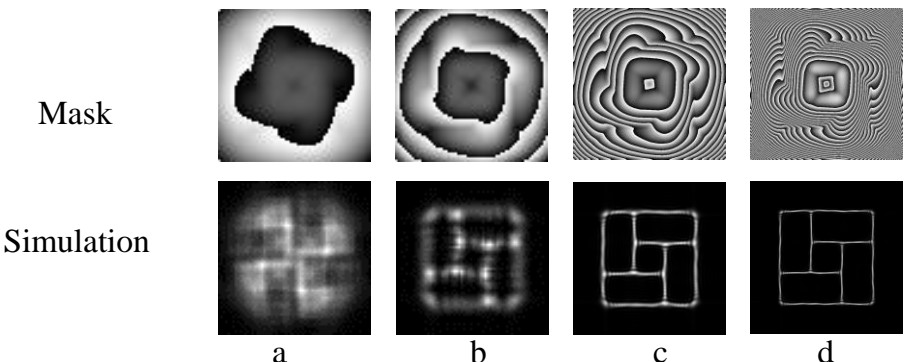

**Figure 2.** Rectangular holographic masks (**top**) and their simulation diffraction pattern (**bottom**) with different $f$ values, where, (**a**,**b**) with $f$ = 5 cm and $f$ = 2cm, and (**c**,**d**) for an expansion mask with $f$ = 5 cm and $f$ = 2cm, respectively.

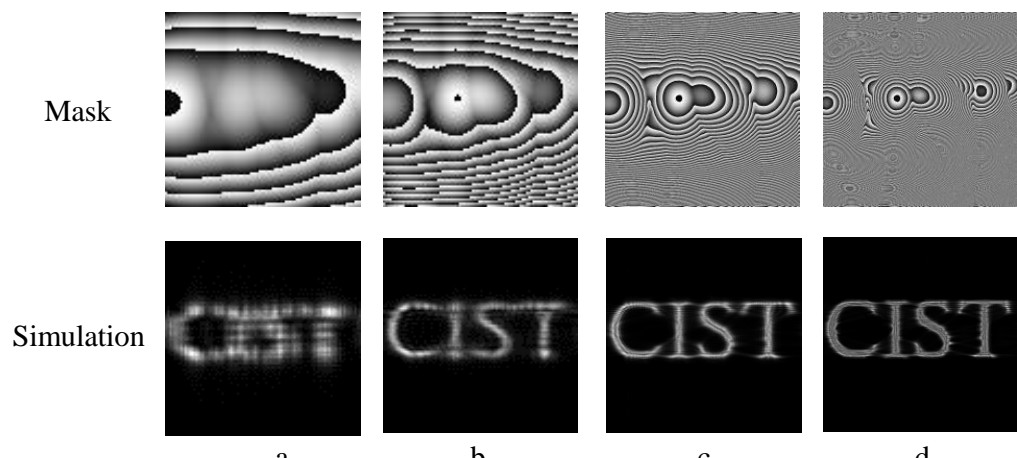

**Figure 3.** Holographic masks of the characters "CIST" (**top**) and their simulation diffraction pattern (**bottom**) with different *f* values, where, (**a**,**b**) with *f* = 5 cm and *f* = 2 cm, and (**c**,**d**) for an expansion mask with *f* = 5 cm and *f* = 2 cm, respectively.

The most straightforward solution to this problem would have been to increase the value of *N*, so that the masks contained more pixels, which would enrich the phase details. However, as we used the Hungarian algorithm, a $100 \times 100$-pixel phase map was the limit for calculations on a desktop computer. Instead, as the phase masks were locally smooth freeform surfaces, we directly interpolated and enlarged them. After expanding the mask heights, lengths and widths 10 times, we obtained the results shown in Figure 2c,d and Figure 3c,d, where the actual $f = 5 \times 10$ cm in Figures 2c and 3c and the actual $f = 2 \times 10$ cm in Figures 2d and 3d. This demonstrated good consistency between the simulation results and the design objectives.

Figure 4 compares the simulated diffraction of the rectangular expanded mask and the target irradiance (in row 320 in the horizontal direction) for various *f* values. Similarly, Figure 5 compares the simulated diffraction of the "CIST" expanded mask and the target irradiance (in row 450 in the horizontal direction).

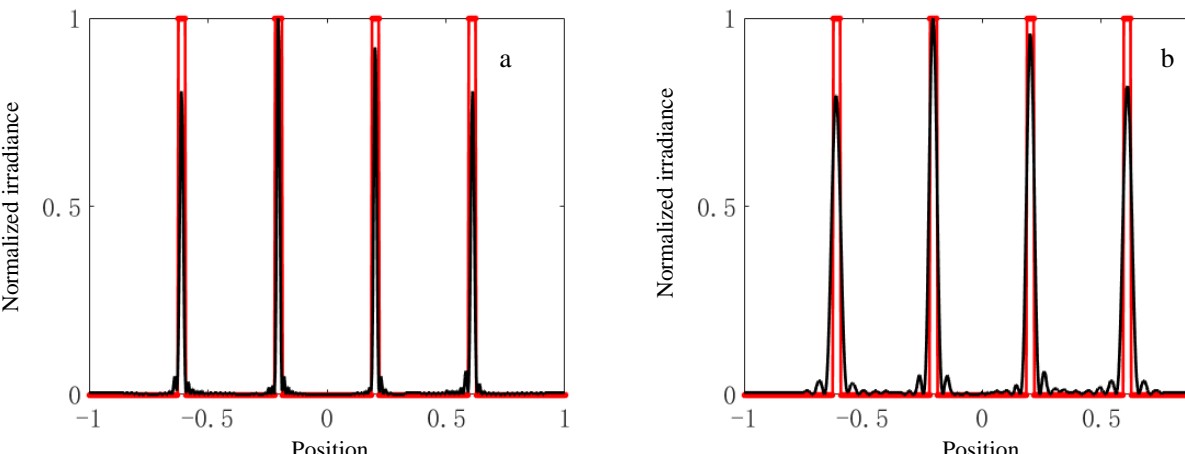

**Figure 4.** Comparison of the simulated (black line) and target irradiance (red line) of the diffraction in-tensity distribution of the rectangular expansion mask (line 320th in the horizontal direction of diffraction pattern) where (**a**) *f* = 2 cm and (**b**) *f* = 5 cm.

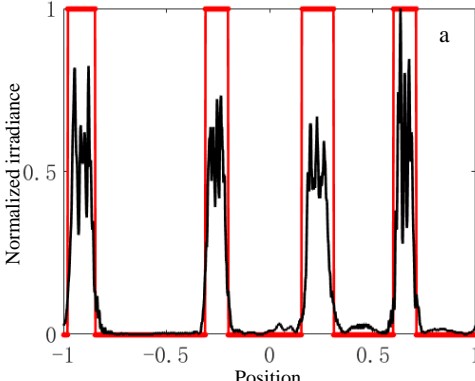 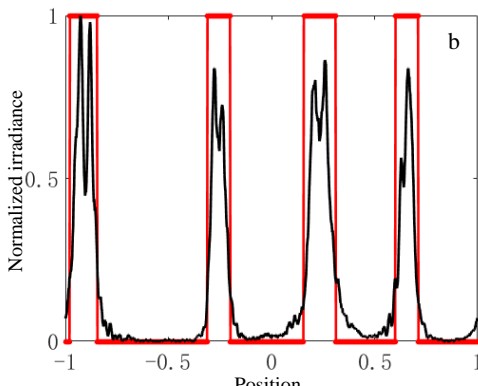

**Figure 5.** Comparison of the simulated (black line) and target irradiance (red line) of the diffraction intensity distribution of the "CIST" expansion mask (line 450th in the horizontal direction of diffraction pattern) with different f value, where (**a**) *f* = 2 cm and (**b**) *f* = 5 cm.

The quantities in Table 1 were calculated using the value of the black line ($I_b$) and the value of the red line ($I_r$) as shown in Figures 4 and 5, as these values reflect the fitting accuracy, i.e., the overall signal-to-noise ratio of the phase distribution $SNR = \sum_n I_r(n) / \sum_n |I_b(n) - I_r(n)|$. The larger the SNR, the higher the matching accuracy of theoretical values and experimental or simulated values. $R1 = 1 - \sqrt{\sum_n (I_b(n) - I_r(n))^2 / \sum_n I_b^2(n)}$ and $R2 = 1 - \sqrt{\sum_n (I_b(n) - I_r(n))^2 / \sum_n I_r^2(n)}$ represent the deviation. The mean absolute value error $MAE = \sum_n |I_b(n) - I_r(n)| / N_l$ and the root mean square error $RMSE = \sqrt{\sum_n (I_b(n) - I_r(n))^2 / N}$ reflect the difference between a theoretical value and an experimental or simulated value. We found that a smaller *f* value did not increase the diffraction exhibited by an expanded mask. In some places with large slopes in the "CIST" mask expansion map for *f* = 2 cm, a phase change of $2\pi$ was only controlled by one or two pixels. This obviously caused errors and clearly indicated a correspondence between phase curvature and diffraction effects, as discussed below.

**Table 1.** Analysis of fitting errors.

|  | SNR | MAE | RNew1 | RNew2 | RMSE |
|---|---|---|---|---|---|
| R_0.02 | 1.6444 | 0.0380 | −0.1276 | 0.3793 | 0.1552 |
| R_0.05 | 1.4472 | 0.0432 | 0.4933 | 0.5595 | 0.1101 |
| CIST_0.02 | 1.6748 | 0.1526 | −0.2070 | 0.4083 | 0.2991 |
| CIST_0.05 | 1.8275 | 0.1398 | 0.1212 | 0.4805 | 0.2626 |

The global mean absolute error (GMAE) was calculated to evaluate the quality of diffraction, where $GMAE = \sum_{n=1}^N |I_b(n) - I_r(n)| / N$. The lower the value of GMAE, the closer a simulation result is to the theoretical value. To simplify the transformation relationship between the value of *f* and the hologram phase curvature, we used *l* = Δh/π as the variable and examined the correspondence between *f* and the hologram phase curvature, where Δh is the height of a mask in the unwrapped state, which is equivalent to the lens sag of the mask. The distribution curves of GMAE under different *l* conditions are shown in Figure 6. It can be seen that an optimal value of *l* made the simulation result closest to the target irradiance value. For example, Figure 6a shows that there was an optimal solution when *l* was approximately 50. Thus, given that the rectangular pattern mask consisted of 640 × 640 pixels, approximately 13 pixels expressed the phase change of $2\pi$ in this case. In contrast, Figure 6b shows that there was an optimal solution when *l* was approximately 130. Thus, given that the "CIST" mask consisted of 900 × 900 pixels, approximately seven

pixels expressed the phase change of $2\pi$ in this case. These two sets of results show that the value of $l$ in the optimal case varied depending on the complexity of the image.

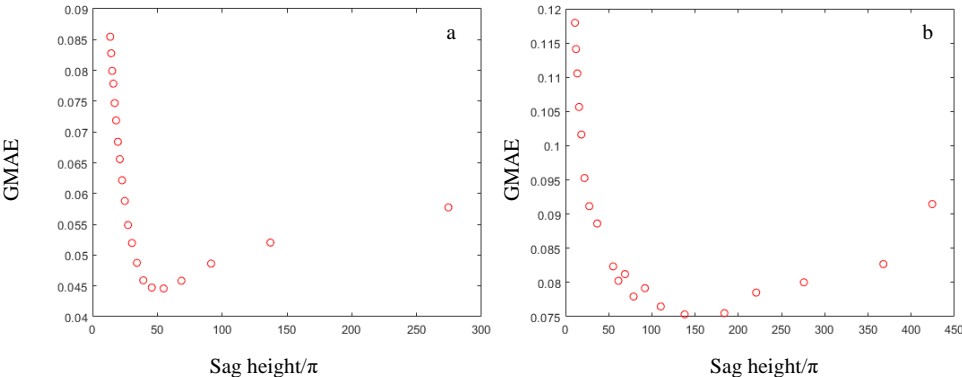

**Figure 6.** Analysis of global mean absolute value error (GMAE) of different sag heights for a rectangular mask (**a**) and characters "CIST" mask (**b**).

### 3.2. Non-Uniform Illumination Focusing

As an example of non-uniform illumination, we computed a phase holographic mask for an image of Einstein's face. The illumination distribution for this image is shown in Figure 7a. The image consisted of 90 × 90 pixels and its number of effective pixels was 4223, which contained four levels of gray. Among them, there are 2122 pixels with the first level of gray, 914 pixels with the second level of gray, 648 pixels with the third level of gray and 539 pixels with the fourth level of gray. These gray values correspond to the light intensity distribution. The mask for this image consisted of 90 × 90 pixels, and the calculated phase mask is shown in Figure 7b. Figure 7c,d show the intensity distribution and the phase distribution of the simulated diffraction pattern for the mask, respectively.

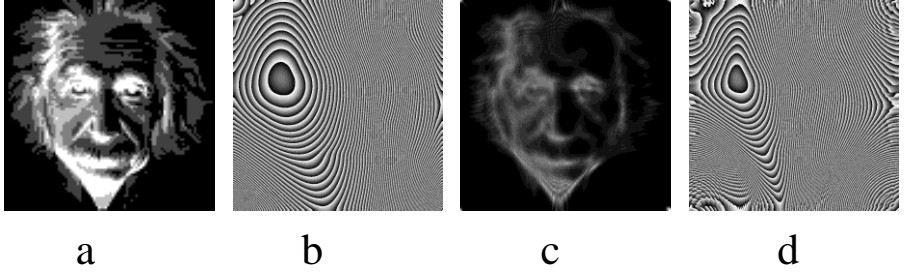

a          b          c          d

**Figure 7.** Example of a non-uniform irradiance mask design and its diffraction spectrum. (**a**,**b**) The original image and its phase-only hologram, respectively, and (**c**,**d**) the amplitude of the simulated diffraction pattern and its phase distribution, respectively.

### 3.3. Illumination Experiment Results

An experimental system containing an SLM was used to observe and verify the diffraction of masks (Figure 8). The light source was a helium–neon laser with a wavelength of 633 nm. The phase-only mask data were loaded onto the SLM from a computer, and the diffraction of the mask was recorded by a CCD camera through a beam splitter (BS). The SLM used was a reflective liquid-crystal-on-silicon SLM (Hamamatsu) with a pixel size of 12.5 μm and a resolution of 1280 × 1024. The laser beam was adjusted to an appropriate size by a beam extender (BE) and then oriented such that it was incident perpendicular to the SLM by the BS, and finally recorded by the CCD camera. So far, due to the limitation of the computing power of the machine, the size we have adopted is relatively small. Two sizes of 90 × 90 pixels and 64 × 64 pixels are adopted. The resolution of SLM used in the experiment is 1280 × 1024 pixels. Since the obtained phase hologram is piecewise smooth, it can be appropriately enlarged to facilitate the diffraction of pixels that can utilize SLM

more, but the enlarged mask will have errors on the boundary of piecewise continuous. Nonetheless, we have found that, in a trade-off between the two, proper magnification improves diffraction. In addition, we verify the diffraction effect of the designed phase-only hologram on SLM. At this time, if the phase hologram is directly loaded on the SLM, similar to the Fresnel lens, other diffraction orders will be mixed into it. In order to remove these influences, we add blazed grating to phase holography, and put lenses L1 and L2 in front of the CCD. Using a diaphragm between L1 and L2 can effectively eliminate the influence of other diffraction orders on the experiment. The results are shown in Figure 9. The expanded mask of $f$ = 5 cm was used, and the results are shown in Figure 9. Figure 9a–c are the experimental diffraction patterns of the rectangle mask, "CIST" mask and Einstein mask, respectively, while Figure 9d is the negative film of Figure 9c. It can be seen that the experimental results are in good agreement with the simulation results. As a result, the existing errors in the experiment are caused by the quantization gray level of SLM itself and the resolution of SLM. If a three-dimensional DOE is made, these influences will be improved. As shown in Figures 2, 3 and 7, the effect of numerical simulation diffraction is very close to the target irradiance (or the original image). However, from the experimental results on SLM, the noise is still relatively large. As we know, the resolution of SLM depends on the size of pixels, which is generally around 10 microns. Of course, due to the continuous progress of technology, the pixel size of the more precise SLM has reached about 5 microns. Even so, as a digital signal, the diffraction effect on the SLM is difficult to compare with the analog signal. We believe that with the advancement of processing technology, the control of light waves using SLM will certainly achieve the modulation effect we expect in the near future.

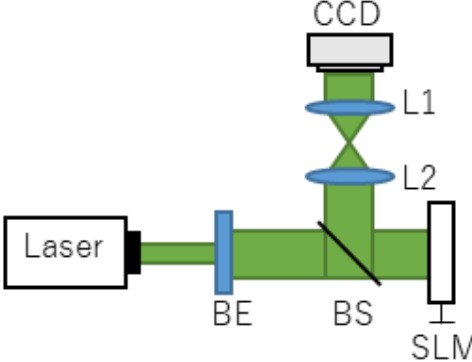

**Figure 8.** Schematic of the experimental system used for holographic recording. SLM: reflective liquid crystal spatial light modulator with pixel size of 12.5 μm and resolution of 1280 × 1024; Laser: He-NE laser with a wavelength of 633 nm; BS: beam splitter; BE: beam expander; L1 and L2: lens with a focal length of 50 mm; CCD = charge-coupled device camera.

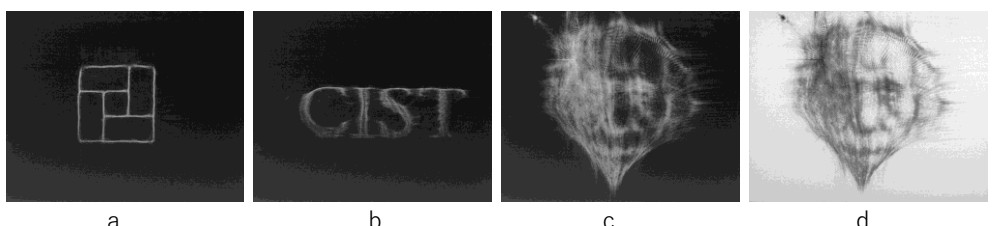

**Figure 9.** Experimentally obtained diffraction patterns of phase-only holographic masks. From (**a**–**c**), they are the diffraction patterns of the rectangular mask, the "CIST" mask and the Einstein image mask, respectively, and (**d**) is the negative of (**c**).

## 4. Conclusions

In this study, we developed a method for calculating phase-only holographic masks. In this method, the mass transmission problem of illumination mapping is transformed into

an LAP, which is solved to obtain the discrete gradient of a phase mask. As the iterative gradient integration method is time-consuming and tends to introduce errors, we used the least squares method devised by Southwell to solve the gradient integral of the phase distribution. To verify the performance of the method, we constructed two holographic masks whose diffraction intensity distributions were a geometric pattern consisting of four rectangles with rotational symmetry and a set of characters with no symmetry ("CIST").

The results showed that the obtained masks exhibited good diffraction effects even after being expanded, which confirmed their piecewise smooth properties. In fact, their diffractive free surfaces were continuous in three dimensions, which is a highly convenient feature for applications in processing, forming and manufacturing. In various scenarios, such as in making DOEs, the phase functions obtained using this method were more convenient than those obtained using phase-only design methods such as the GS algorithm or the BERD algorithm. Moreover, the calculation time of this method was approximately 10 min, whereas that of the SQM is typically 1–2 h. Furthermore, our method has no specific requirements for the symmetry and continuity of a target light-intensity distribution. We also compared the mask simulations with the target light-intensity distributions. For a given mask size, there was an optimal value of lens sag that minimized the GMAE value. We found that when selecting the optimal lens sag value for mask design, the rectangular mask used approximately 13 pixels to express a phase change of $2\pi$, whereas the "CIST" mask used approximately 7 pixels to express a phase change of $2\pi$. As mentioned above, the method proposed in this paper is an effective method for freeform surface design. The resulting mask mold has piecewise smoothing property, that is, the phase transition is continuous. In addition to amplitude control, this kind of method may also have great application prospects in phase control.

In this paper, we successfully use the LAP method to realize the design of phase-only holography in the focusing imaging process of non-uniform illumination intensity distribution. For the non-uniform illumination image, we used an SLM to verify its diffraction. The results showed that using the phase-only mask formed in the experiment effectively reproduced the diffraction pattern of the target image.

**Author Contributions:** Methodology, K.S. and G.Z.; software, K.S.; validation, K.S. and G.Z.; formal analysis, K.S.; investigation, G.Z.; data curation, G.Z.; writing original draft preparation, K.S.; writing review and editing, G.Z. All authors have read and agreed to the published version of the manuscript.

**Funding:** This research received no external funding.

**Institutional Review Board Statement:** Not applicable.

**Informed Consent Statement:** Not applicable.

**Data Availability Statement:** Not applicable.

**Conflicts of Interest:** The authors declare no conflict of interest.

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
