# Peer review of "Design and Application of Phase-Only Diffractive Optical Element Based on Non-Iterative Method"

_photonics, doi:10.3390/photonics9110874_

Round 1
Reviewer 1 Report
The paper discusses an approach for calculating phase masks that can generate arbitrary intensity distributions. The paper as a whole has a very extensive theoretical/design section and a brief experimental one. There are several major issues that the authors need to address and fix in order for this paper to be of interest to the general public.
1. Lines 32-35 - the authors reject the IFTA method as not good for creating phase masks. That is an approach used and improved for tens of years that has demonstrated close to perfect results. So, the statement done in lines 32-35 is not correct and is not even clear what exactly it means. DOEs made by this approach "does not retain phase information"? What exactly does this means?
2. The authors need to clarify what exactly they mean by "focus length of a mask" - line 182. Is focusing information included in the phase mask by design? If so, can this be avoided? Regular phase masks do not include focusing properties unless that was specifically required and correspondingly included into the design.
3. All the simulated intensity and phase patterns are 90x90 pixels. In the same time the experiments use 1280x1024 pixel SLM. Why such choice was made and is that the reason for the poor experimental results?
4. The experimental results presented are not very close to the theoretical ones. There is a lot of noise, smearing and lack of edge definition (Fig. 9). The authors need to address all these and redo their experiments so much better results are achieved. Without better experimental results the paper is not worth publishing given that the authors' intent is to show comparable and even better than the existing methods for calculating CGHs.
5. In section 3.4. the authors are trying to make an OAM beam. It is not clear why there is a need to impose the LAP method phase pattern on top of the Vortex phase one? The vortex one is totally sufficient to produce OAM beam. So, what is the purpose and why make the complication of adding extra phase?
6. May be the authors are not aware but there are very simple methods for testing the presence of helical phase in an OAM beam. Pure interference with a collimated reference beam will create a fork grating pattern and interference with a divergent beam will create a spiral one. So, I do not see the purpose of the complex scheme presented by the authors.
As a whole section 3.4. does not present any results that might be worth publishing.
It is this reviewer's opinion that the paper has major issues and needs a lot of work in order to be published.
Author Response
Dear reviewers,
Thank you very much for your valuable comments on our manuscript. There is no doubt that these comments are valuable and very helpful for revising and improving our manuscript. In what follows, we would like to answer the questions you mentioned and give detailed account of the changes made to the original manuscript.
Yours sincerely,
Gongjian Zhang and Kuo Shi
……………………………………………………..
Chitose Institute of Science and Technology,
758-65,Bibi, Chitose, Hokkaido,
066-8655, Japan,
Phone/Fax: +81-123-27-6118
Emails: zhang@photon.chitose.ac.jp
shikuo@live.cn,
……………………………………………………..
Comment 1
Lines 32-35 - the authors reject the IFTA method as not good for creating phase masks. That is an approach used and improved for tens of years that has demonstrated close to perfect results. So, the statement done in lines 32-35 is not correct and is not even clear what exactly it means. DOEs made by this approach "does not retain phase information"? What exactly does this means?
Response:
We gratefully appreciate for your valuable comment.
As a mature method to manufacture phase mask, IFTA method has been widely used in practice. We rewrote the introduction part of the article to make the description clear and avoid ambiguity.
p.1
In lines 30-36 description
“However, phase-only holograms generated by current methods, such as the Gerchberg–Saxton (GS) algorithm [7] and the bidirectional error diffusion (BERD) method [8], tend to have rough surfaces as they lack continuous differentiability, i.e., piecewise smoothness. These rough-surfaced pure-phase holograms are difficult to employ in the manufacturing of diffractive optical elements (DOEs) and do not effectively retain phase information.”
was rewritten as follows:
“At present, iterative Fourier transform algorithms(IFTA) methods are widely used in the generation of phase-only holography and have achieved great success [7,8]. However, the diffractive optical element (DOE) micro-reliefs obtained by it are usually complex and irregular in shape, often similar to white noise. Such relief is difficult to manufacture, and the manufacturing error leads to the uncontrolled scattering of light on the micro relief and the formation of speckle structure, which greatly damages the quality of the generated light irradiation distribution.”
Comment 2
The authors need to clarify what exactly they mean by "focus length of a mask" - line 182. Is focusing information included in the phase mask by design? If so, can this be avoided? Regular phase masks do not include focusing properties unless that was specifically required and correspondingly included into the design.
Response:
Thank you for your valuable comment. To avoid ambiguity, we refer to the DOE imaging distance as "focus length of a mask" in this paper. According to the suggestions of reviewers, in the line 160(p.4), the following description is added:
“As described in reference 13, the process of ray mapping can be seen as a process of focusing a beam to a specified area. And the focal plane is the plane z equals f. Therefore, the propagation distance f is also described as the focal length of mask in the later description of this paper.”
Comment 3
All the simulated intensity and phase patterns are 90x90 pixels. In the same time the experiments use 1280x1024 pixel SLM. Why such choice was made and is that the reason for the poor experimental results?
Response:
For ease of design, we used a phase mask size of 90x90 pixels. This is described in the article. The computational amount of designing large-size holographic mask is difficult to be realized on the current computer. See (see line 203 in the article). The SLM used in the lab, on the other hand, has a resolution of 1280x1024 pixels. Therefore, this choice appears. The main reason for the poor experimental results is the small size of DOE and the diffraction of other orders mixed in the SLM verification of experimental results. We take into account that although the expansion of MASK will cause errors on the piecewise continuous boundary, larger mask is beneficial to SLM diffraction, so proper expansion can improve the diffraction effect. For this reason, we also made a further description in the article.
p.8
In lines 289, the following a section of discussion added:
“So far, due to the limitation of the computing power of the machine, the size we have adopted is relatively small. Two sizes of 90x90 pixels and 64x64 pixels are adopted. The resolution of SLM used in the experiment is 1280x1024 pixels. Since the obtained phase hologram is piecewise smooth, it can be appropriately enlarged to facilitate the diffraction of pixels that can utilize SLM more, but the enlarged mask will have errors on the boundary of piecewise continuous. Nonetheless, we have found that, in a trade-off between the two, proper magnification improves diffraction. In addition, we verify the diffraction effect of the designed phase-only hologram on SLM. At this time, if the phase hologram is directly loaded on the SLM, similar to the Fresnel lens, other diffraction orders will be mixed into it. In order to remove these influences, we add blazed grating to phase holography, and put lenses L1 and L2 in front of CCD. Using a diaphragm between L1 and L2 can effectively eliminate the influence of other diffraction orders on the experiment. The results are shown in Fig.9.”
Comment 4
The experimental results presented are not very close to the theoretical ones. There is a lot of noise, smearing and lack of edge definition (Fig. 9). The authors need to address all these and redo their experiments so much better results are achieved. Without better experimental results the paper is not worth publishing given that the authors' intent is to show comparable and even better than the existing methods for calculating CGHs.
Response:
We gratefully thank you for the precious time the reviewer spent making constructive remarks. The experimental result is different from the theory. We have made further investigation, which is due to the influence of other diffraction orders. Therefore, we have improved the experimental method in 3.3. Figure 9 is remeasured result.
p.9
Above Fig. 9, the following paragraph is added immediately.
“As a result, the existing errors in the experiment are caused by the quantization gray level of SLM itself and the resolution of SLM. If three dimensions DOE is made, these influences will be improved. As shown in Fig. 2, 3 and 7, the effect of numerical simulation diffraction is very close to the target irradiance (or the original image). However, from the experimental results on SLM, the noise is still relatively large. As we know, the resolution of SLM depends on the size of pixels, which is generally around 10 microns. Of course, due to the continuous progress of technology, the pixel size of the more precise SLM has reached about 5 microns. Even so, as a digital signal, the diffraction effect on the SLM is difficult to compare with the analog signal. We believe that with the advancement of processing technology, the control of light waves using SLM will certainly achieve the modulation effect we expect in the near future.”
Comment 5
In section 3.4. the authors are trying to make an OAM beam. It is not clear why there is a need to impose the LAP method phase pattern on top of the Vortex phase one? The vortex one is totally sufficient to produce OAM beam. So, what is the purpose and why make the complication of adding extra phase?
Response:
We really appreciate your valuable advice. Considering that this method can be used to design any beam with vortex phase intensity distribution. Since the objectives of this study are somewhat scattered, the content of Section 3.4 has been deleted in accordance with your comments.
Comment 6
May be the authors are not aware but there are very simple methods for testing the presence of helical phase in an OAM beam. Pure interference with a collimated reference beam will create a fork grating pattern and interference with a divergent beam will create a spiral one. So, I do not see the purpose of the complex scheme presented by the authors.
Response:
We gratefully appreciate for your valuable suggestion. These methods are simple and effective to verify the vortex phase. In order to verify the phase information of the diffraction beam obtained by experiment, we have made a practical measurement. It can be seen that in the part where the light intensity distribution is not zero. Its phase is consistent with the simulation.
Since the objectives of this study are somewhat scattered, the content of Section 3.4 has been deleted in accordance with your comments.

Reviewer 2 Report
The authors propose and demonstrate the design of continuous phase-only holographic masks that map laser light to arbitrary target illumination patterns. The manuscript is well written, and analytical and experimental details are well documented. However, I have few questions that would be good to see discussed or modified in the final manuscript and, assuming the answers make sense, I believe it is acceptable for publication.
1. In the “Design examples” part, the author points out that Because the focal length f = 5 cm was too small, the curvatures of the masks were very small. The overall details of the masks were thus compressed, which made the diffraction results very fuzzy. Then the author designed a 2cm focal length. According to the author, the smaller the focal length, the worse the effect. Why the diffraction result with f = 2 cm was significantly better than that with f = 5 cm?
2. In Figures 4 and 5, the title of the abscissa is not centered.
3. In 3.2 “Non-uniform illumination focusing” part. Sentence “There were 4223 points with a gray value of 1, 914 points with a gray value of 648 points with a gray value of 3, and 539 points with a gray value of 4. These gray values correspond to the light intensity distribution If.” should be rewritten.
Author Response
Dear reviewer,
Thank you very much for your valuable comments on our manuscript. There is no doubt that these comments are valuable and very helpful for revising and improving our manuscript. In what follows, we would like to answer the questions you mentioned and give detailed account of the changes made to the original manuscript.
Yours sincerely,
Gongjian Zhang and Kuo Shi
……………………………………………………..
Chitose Institute of Science and Technology,
758-65,Bibi, Chitose, Hokkaido,
066-8655, Japan,
Phone/Fax: +81-123-27-6118
Emails: zhang@photon.chitose.ac.jp
shikuo@live.cn,
……………………………………………………..
Comment 1
In the “Design examples” part, the author points out that Because the focal length f = 5 cm was too small, the curvatures of the masks were very small. The overall details of the masks were thus compressed, which made the diffraction results very fuzzy. Then the author designed a 2cm focal length. According to the author, the smaller the focal length, the worse the effect. Why the diffraction result with f = 2 cm was significantly better than that with f = 5 cm?
Response:
Thank you so much for your careful check. we were really sorry for careless mistakes.
p.5
In lines 191 through 192, “Because the focal length f = 5 cm was too small, the curvatures of the masks were very small.” was corrected as “Because the focal length f = 5 cm was long, the curvatures of the masks were very small.”
Comment 2
In Figures 4 and 5, the title of the abscissa is not centered.
Response:
Thank you for your valuable comment. We have remade Figures 4 and 5 so that their titles are centered.
Comment 3
In 3.2 “Non-uniform illumination focusing” part. Sentence “There were 4223 points with a gray value of 1, 914 points with a gray value of 648 points with a gray value of 3, and 539 points with a gray value of 4. These gray values correspond to the light intensity distribution If.” should be rewritten.
Response:
Thank you for your valuable comment. We have adjusted the text to make it clearer and clearer.
p.8
Rewrite lines 265 to 269 as
“The image consisted of 90 × 90 pixels and its number of effective pixels was 4223, which contained four levels of gray. Among them, there are 2122 pixels with the first level of gray, 914 pixels with the second level of gray, 648 pixels with the third level of gray and 539 pixels with the fourth level of gray. And these gray values correspond to the light intensity distribution.”

Round 2
Reviewer 1 Report
Thank you for taking into account the remarks and for making the changes.
The paper now is in a much better shape.